# Candidate Genes in Bull Semen Production Traits: An Information Approach Review

**DOI:** 10.3390/vetsci9040155

**Published:** 2022-03-23

**Authors:** Mamokoma Cathrine Modiba, Khathutshelo Agree Nephawe, Khanyisile Hadebe Mdladla, Wenfa Lu, Bohani Mtileni

**Affiliations:** 1College of Animal Sciences and Technology, Jilin Agricultural University, Xincheng Street, Changchun 130118, China; 205211926@tut4life.ac.za; 2Joint Laboratory of Modern Agriculture Technology International Cooperation Ministry of Education, Jilin Agricultural University, Changchun 130118, China; 3Department of Animal Sciences, Tshwane University of Technology, Private Bag X680, Pretoria 0001, South Africa; nephaweka@tut.ac.za; 4Agricultural Research Council Biotechnology Platform, Private Bag X5, Onderstepoort, Pretoria 0110, South Africa; mdladlak@arc.agric.za

**Keywords:** semen production parameters, candidate genes, genome-wide association studies, bull

## Abstract

Semen quality plays a crucial role in the successful implementation of breeding programs, especially where artificial insemination (AI) is practiced. Bulls with good semen traits have good fertility and can produce a volume of high semen per ejaculation. The aim of this review is to use an information approach to highlight candidate genes and their relation to bull semen production traits. The use of genome-wide association studies (GWAS) has been demonstrated to be successful in identifying genomic regions and individual variations associated with production traits. Studies have reported over 40 genes associated with semen traits using Illumina BeadChip single-nucleotide polymorphism (SNPs).

## 1. Introduction

Semen traits are important for cattle gene improvement programs relying on artificial insemination [1]. The collection of bull semen and the scoring of parameters has become the standard routine in many AI centers [2]. Semen production traits that are routinely assessed from fresh ejaculates include semen volume (VOL), sperm concentration (SCONC), sperm motility (SM), and the proportion of sperm with a head (PSH). These traits have key value in assessing semen in AI companies and are also essential when selecting the top-performing bulls [3]. Young bulls are used in AI programs for a short time (4–6 months), which again highlights the importance of semen production traits [3].

Studies [4,5] have focused on identifying genes and genetic markers associated with bovine semen traits in order to understand their genetic architectures. A number of studies on candidate genes associated with semen quality traits have been conducted in pigs [6,7], horses [8], sheep [9], and goats [10,11]. However, the population sizes analyzed in these studies were not large (139–900), with low statistical power to detect causal genes. A small number of genes are shared between these studies; therefore, the genetic structure underlying semen traits remains unknown. Genome-wide association studies can help to overcome the limitations by using single-nucleotide polymorphism (SNP) genotyping assay tools to examine traits of significance [12]. Fortes et al. (2013) [13] revealed that significant SNPs in the X-chromosome were associated with the percentage of progressive motile spermatozoa at 18 months of age and the percentage of normal spermatozoa at 24 months of age in tropical composite bulls. A study conducted by Gottschalk et al. (2016) [8] on 139 German Warmblood hors.es identified that 29 SNPs on 12 unique chromosomes were associated with semen quality traits. A number of studies have identified candidate genes in livestock and continue to demonstrate the efficiency of GWASs to identify semen traits [14,15,16]. Hering et al. (2014) [14] conducted GWASs on 41 bulls with very poor sperm motility and 279 bulls with excellent sperm motility and identified nine candidate genes all with a strong relationship to sperm function. Hence, this review uses the information approach to highlight candidate genes, and their relation to bull semen production traits.

## 2. Association Studies for Semen Traits

A genome-wide association study (GWAS) is an approach used in genomic research to associate specific genotype variations with a particular phenotypic trait. Genome-wide association studies evaluate genomes from several phenotypes, looking for genetic markers that can be used to predict the presence of a trait. Once such markers are identified, they can be used to understand how genes contribute to the traits. Genome-wide association studies have several factors that contribute to their successful utility. One example is the phenotypic variation; filtering phenotype data is important to avoid outliers during analysis. The population size is very important for obtaining meaningful results for both phenotypic and genotypic variation; one example is a recent study on 1819 Angus bulls with 50,624 records for a single-step genome-wide association study (ssGWAS) on the following traits: VOL, SCONC, number of spermatozoa (NS), initial motility (IMOT), post-thaw motility (PTMOT), three-hour post-thaw motility (3HRPTMot), percentage of normal spermatozoa (%NORM), primary abnormalities (PRIM), and secondary abnormalities (SECs) [16]. Their findings indicated regions of the genome that impacted fertility and included genomic information in the genetic evaluation, which is advantageous for genetically improving male fertility traits [16]. Single-nucleotide polymorphisms (SNPs) are bi-allelic genetic markers, easy to evaluate and interpret, and widely distributed within genomes. These SNPs can compute linkage disequilibrium (LD), fixed information in the genome used to classify fundamental genes of adaptation in domestic animals. Genome-wide studies have been conducted in both beef and dairy cattle and have revealed several genomic regions associated with semen traits.

Hering et al. (2014) [14] reported markers in Holstein Friesian bulls that were significant for semen volumes on BTA 10 and 22: rs42438348, rs41625599, rs41584616, and rs42012507 and the total number of motilities on BTA 22 rs41625599, rs41584616, rs42012507, and rs110109069. They further indicated that only 19 SNPs were significantly associated with five semen traits. Recently, studies of association have depended on the genotyping of thousands of SNP markers to evaluate genomic breeding value. One study mapped 34 SNPs in 19 different Bos taurus autosomes and one on chromosome X: significant SNPs were located on chromosome 24 (rs110876480), 5 (rs110827324 and rs29011704), and 1 (rs110596818), in close vicinity to the melanocortin 4 receptor (MC4R) gene, the PDZ domain containing the ring finger 4 (PDZRN4) gene, the ethanolamine kinase 1 (ETNK1) gene, and the olfactory receptor 5K3-like (LOC785875) gene, respectively [15]. The cost of processing GWASs is reasonable, due to the low cost of sequencing and the ability to determine large datasets. GWASs require a bi-allelic assumption, which is reasonable because only 1–3% of the genome has random copy number variants. Liu et al. (2017) [17] reported that one SNP (rs110305039) located on downstream of PDGFRB was significantly related to semen volume per ejaculate (SVPE), the number of sperm per ejaculate (NSPE), and the number of motile sperm per ejaculate (NMSPE). This SNP rs110305039 was previously revealed to have an association with sperm motility (SM). Liu et al. (2017) [17] further revealed three significant SNPs, rs211260176, rs208093284, and rs43445726, located on the promoter of MARCH1, which showed associations with the semen volume per ejaculate (SVPE), number of sperm per ejaculate (NSPE), and number of motile sperm per ejaculate (NMSPE), respectively. Figure 1 shows the distribution of the 13642458 bp length start and 19755465 bp length end SNP on the chromosome, which was taken from our GWAS data using QTL/association. Recently, Butler et al. (2022) [18] reported five SNPs, BTB-01549373, rs41666488, rs109736826, rs109268478, and rs41575945, which are strongly associated with SV, and three significant SNPs, rs43067163, rs41623602, and rs29023737, which were identified for influencing concentration (CONC) [18]. Moreover, for IMot, six SNPs on six different chromosomes were significant, including rs41623436, rs109798673, rs43526428, rs29003479, rs42861585, and rs109512383 [18]. For % NORM, six significant SNP were identified: rs41606310 on chromosome one, rs110964837 on chromosome two, rs41594758 on chromosome three rs109928164 on chromosome five, rs41591913 on chromosome five, and rs41666416 on chromosome ten [18].

### 2.1. Single-Nucleotide Polymorphism Markers Used to Identify Associations

Genome-wide association studies have relied on certain types of single-nucleotide polymorphism markers to obtain information in different populations and species. The SNPs selected in GWASs are either Illumina (San Diego, CA, USA) or Affymetrix (Santa Clara, CA, USA). These two contending technologies have been selected and reviewed and bring unique approaches to measure SNP differences. Affymetrix (chip-based) displays short DNA sequences to detect alleles, whereas Illumina (bead-based) displays slightly longer DNA sequences to detect alleles. A study by Suchocki and Szyda (2015) [4] reported that, recently, most genome-wide association studies on fertility traits have been conducted using the bovine 50 K Illumina SNP chip. The information gathered by these markers can assist in identifying phenotypic differences and highlighting the accuracy of animal selection for mating purposes, thus increasing genetic gains for advanced selection pressures [19]. However, numerous association studies conducted on complex traits have used Illumina BeadChip SNPs, especially in studies identifying semen production traits. A summary of studies conducted using Illumina BeadChip SNPs is presented in Table 1.

### 2.2. Candidate Genes Associated with Semen Traits

The candidate gene approach is useful for quickly determining the associations of a specific genetic variant with a phenotype and the number of relevant genes governing traits [23]. To date, several candidate genes have been detected that influence valuable traits, even though the total amount of published information of putative genes remains quite small. The candidate genes approach has been criticized due to the low reputability of results and the limitations of including relevant genes [24]. The challenge is that it requires existing information on physiological, biochemical, or functional knowledge and the biochemical metabolism pathway, which may not be available [25]. Candidate gene association studies evaluate genetic variations associated with traits within a limited number of pre-specified genes [26]. For instance, candidate gene studies have more significant outputs in identifying variation, critical genes, and biological pathways [26].

Many putative candidate genes in both dairy and beef cattle have been identified to be associated with bull semen traits. Studies on genes of biological pathways showed an associated with semen traits, which revealed seven significant biological pathways relating to 127 genes with 1.04% of the genetic variation for the volume (VOL), number of sperm (NS), and motility (MOT) [27]. Nonetheless, studies on candidate genes have been performed, and several genetic variations have been identified in bull semen. Borowska et al. (2018) [28] reported that candidate genes had two shared SNPs related to sperm plasma integrity and used GWASs for SC records on 392 bulls and identified the STATU2 gene which participates in multiple biological processes, including reproduction and developmental and immune structures. Hering, Olenski, and Kaminski (2014) [3] used two approaches to find candidate genes potentially related to very poor sperm motility: (1) a physical approach, which involved genes in the vicinity of a significant pathway, an approach that identified nine candidate genes positioned close to significant SNP markers found to have a strong relationship with sperm function (LOC785875, ALPL, HIBADH, GADD45G, TRIM36, TGFA, KLHL1, PRKCB, and INCENP); (2) a functional approach, which is the identification of candidate genes with a close distance of 1 mb. These strategies can assist in identifying candidate genes associated with important traits of interest. The genes identified above support the hypothesis that looking for candidate genes only among those in the direct vicinity of significant markers can overlook other genes. Table 2 shows the genes found in several studies of BTA positions and SNPs of interest. Genes such as FSHR, INHA, TNP1, TNP2, CAPN1, and SPAG11 have been studied as candidate genes for influencing semen traits in bulls [25,29], as shown in Table 2 and Table 3. The benefits of sequencing a whole bovine genome [25] with many SNPs can simplify the localization of specific gene regions associated with traits of importance. A study on genomic regions related to the inbreeding depression of live spermatozoa identified 53 genes, and an additional analysis of efficient annotations of the genes recognized nine strong candidate genes related to male fertility, located on chromosomes 1, 6, 10, and 14 [30]. However, there appears to have been surprisingly little work conducted to date to characterize bovine Y chromosome genes for male fertility [27]. This is probably because, until recently, there has been no reference sequence assembly for the bovine Y chromosome [27]. One candidate gene commonly found to be associated with bull sperm quality is the SOX5 (SRY-box5) gene, which encodes for the sex-determining Y chromosome box5 protein [28]. Five of the prioritized candidate genes for sperm motility have also been formerly revealed as candidate genes in the regulation of male fertility. These include superoxide dismutase 2, T-complex protein 1, parkin co-regulated gene, sperm flagella 2 gene, and prolactin receptor (SOD2, TCP1, ACRG, SPEF2, and PRLR). One study reported on 22 novel candidate genes which were identified on chromosomes 1, 5, 6, 7, 15, 17, 23, and 27 [5]. There are many more window regions affecting traits, which explain up to 1% of genetic variance. Results from Qin et al. (2017) [5] exhibited candidate genes PDE3A and SLCO1C1, where PDE3A was on BTA5. Candidate gene studies can overcome these issues, focusing directly on the association between disease and variants in particular genes that have a priori biological support. This focus comes at a cost: candidate gene studies ignore much of the genome and thus are likely to miss many causal regions or genes and instead find many false-positive associations.

### 2.3. Genes Detected in Dairy and Beef Cattle

The identification of candidate genes provides a better understanding of the distribution of genes that affect traits of economic interest [48]. Hering et al. (2014) [3] reported on the role of the following genes associated with sperm motility using a physical approach in a Holstein population: MC4R, ETNK1, LOC785875, TRIM36, ALPL, PRKCB, HIBADH, KHL1, PD2RN4, CTTR, SRD5A2, CAPN1, CATSPER1, ATP5O GABRR3 EIF4G3, SOX5 PTPRB PTPRR, SECISBP2, CYLC2, CRYZL1, LOC785875 ZBTB40, ALPL, AGBL4 ST7, PDZRN4 CNOT2, ETNK1 MRPL1, RAPGEF6, FREM1 GADD45G SPIN1 GRIN3A TRIM36 LOC511898 TGFA, FAM84B, LOC100139627, CYP2C87, SORCS1, LOC101905219, and DLC1 (shown in Table 4). In another study, Hering et al. (2014) [14] highlighted genes DCP1, SFMBT1, GALC, PRKCD, PHF7, TLR9, SPATA7, and TMEM110 associated with semen volume and the total number of sperm in Holstein Friesian bulls, located in the vicinity of significant markers. However, most studies on candidate gene association have strongly focused on dairy cattle, as shown in Table 4. For beef cattle, however, there is limited information on candidate gene association studies. Sweett et al. (2020) [16] identified five candidate genes in 265 crossbred beef bulls for SM in the regulation of male fertility. These genes include superoxide dismutase 2, T-complex protein 1, parkin co-regulated gene, sperm flagella 2 gene, and prolactin receptor (SOD2, TCP1, PACRG, SPEF2, and PRLR). Other results on beef cattle have been documented by Butler et al. (2022) [18]; genes associated with fertility were found to be close to the significant SNPs in the study. There is still a gap where dairy cattle research has capitalized on genomic technologies [3,49] and multiple QTL regions [5,46] and candidate genes [50,51] associated with male and female fertility have been identified. There is limited information on fertility traits in beef cattle bulls [16]. This gap was highlighted by studies conducted by Sweett et al. (2020) [16] and Butler et al. (2022) [18], which showed a lack of attention given to beef cattle regarding candidate genes. Sweett et al. (2020) [16] even stated that SM measurements are much less common in the beef industry, where natural breeding is often used, as can be seen in our sample size.

Some of these genes were reported to have a significant role in spermatogenesis, as shown in Table 4. The genes LOC785875, ALPL, HIBADH, GADD45G, TRIM36, TGFA, KLHL1, PRKCB, and INCENP have a strong relationship with sperm function, which confirms the important role of these genes on semen quality [3]. Table 5 represent all the genes found in GWAS studies and in my review paper. 

## 3. Traits in Most GWAS Studies

Semen traits are easily measurable, and record-keeping for each bull is essential in any AI company. Most of the studies conducted on bull semen traits have focused on several important parameters, such as motility, progressive motility, and morphological abnormalities of sperm [3,17,62]. These are some of the parameters regarded as critical in assessing semen by AI companies. However, selecting animals directly based on their semen phenotypes can be difficult because of the low (0.04) to moderate (0.30) heritability of these traits [49,63]. Gredler et al. (2007) [63] demonstrated the moderate heritability of all semen parameters with a low correlation between breeding values for semen quality traits, and routinely estimated the breeding values for male fertility. Stälhammar et al. (1989) [64] highlighted the low to moderate heritability of semen parameters in Swedish Red, White, and Swedish Friesian bulls. The demand for bulls with good semen parameters is very high; hence, regular production should be a priority for AI companies [30]. Studies in the literature have shown moderate to high heritability for some of the parameters, whereas in other studies, heritability is lower, making it hard to select the same parameters in other animals. Yin et al. (2019) [46] studied the heritability of the following semen parameters: ejaculate volume (VE), progressive sperm motility (SM), sperm concentration (SC), number of sperm (NSP), and number of progressive motile sperm (NMSP) and demonstrated positive correlation amongst all five parameters. An article by Liu et al. (2017) [17] evaluated five semen traits and showed the greatest positive correlation amongst all traits, with the highest correlation observed between the number of sperm per ejaculation and the number of motile sperm per ejaculation. However, there is limited information on some of the parameters in these studies; most of these evaluated parameters are repeated across studies. Karoui et al. (2011) [65] revealed moderate heritability for the volume, concentration, number of spermatozoa per ejaculate, mass motility score, and post-thawing motility traits; only individual motility was low in heritability. A more inclusive view regarding the semen traits to be measured across all parameters could broaden bull selection criteria in the future. This can expand knowledge of the importance of semen production parameters in bulls. Additionally, to improve the power of the analysis, multi-trait techniques and denser marker maps are required for reference for future studies.

## 4. Conclusions

Our review has highlighted genes associated with semen parameters; over 40 genes have been documented in the literature and have been reported by several authors to be associated with semen traits. We have furthered the understanding of how the length of the marker highlights the position of a gene to influence a trait in chromosomes. Marker efficiency is vital in candidate association studies in which Illumina is the most widely used marker. 

## Figures and Tables

**Figure 1 vetsci-09-00155-f001:**
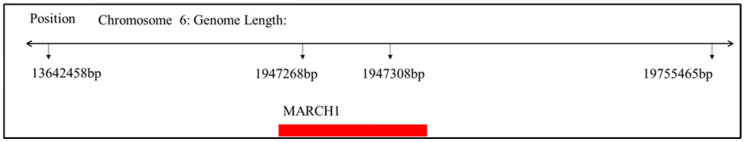
Length of marker position of a gene (MARCH1) and Chromosome 6 (https://www.animalgenome.org/cgi-bin/QTLdb/BT/search accessed on 29 January 2022).

**Table 1 vetsci-09-00155-t001:** Studies conducted using Illumina on semen traits.

Population Size	Target Traits	Snps Used	Articles Published
1581 Holstein Friesian bulls	Sperm motility.	Illumina BovineSNP50KBeadChip, 54,001.	[3]
Polish, Holstein Friesian 1212 bulls	Sperm concentration, semen volume, number of spermatozoa, motility, and motility score.	Illumina BovineSNP50K BeadChip 54,001.	[4]
692 Holstein bulls	Ejaculate volume, sperm concentration, sperm motility, and sperm concentration	Illumina BovineSNP50 BeadChip, SNP markers	[5]
1085 tropical composite bulls	Sperm motility and percentage of normal spermatozoa.	Illumina Bovine HD SNP50 BeadChip	[13]
788, Holstein Friesian bulls	Semen volume and total number of sperm	Illumina BovineSNP50 Bead-Chip, 54,001.	[14]
730 Chinese Holstein bulls	Semen volume per ejaculate, sperm motility, sperm concentration per ejaculate, the number of sperms per ejaculate, and the number of motile sperms per ejaculate.	Illumina Bovine SNP50 BeadChip	[17]
1819 Angus	Volume, concentration, number of spermatozoa, initial motility, post-thaw motility, three-hour post-thaw motility, percentage of normal spermatozoa, primary abnormalities, and secondary abnormalities.	Illumina Bovine HD SNP50 BeadChip	[18]
1799, Austrian Fleckvieh bulls	Total number of spermatozoa and percentage of live spermatozoa.	Illumina Bovine SNP50K BeadChip 54,001.	[20]
1099 Brahman and Tropical bulls 1719	Sperm morphology and sperm chromatin phenotypes.	Illumina BovineSNP50 BeadChip, 54,000.	[21]
2481 Brown Swiss bulls	Ejaculate volume, sperm concentration, sperm motility, sperm head, and tail anomalies	Illumina Bovine HD SNP50 BeadChip 777,962SNPs	[22]

**Table 2 vetsci-09-00155-t002:** Candidate genes, markers, and chromosome positions [3].

SNP Name	Position	Identified Candidate Genes	Reference
rs29010277	X	LOC100848828	[4]
rs110419531	X	*MAGEB10*	[4]
rs109349108	X	*MAGEB10*	[4]
rs110685046	X	*MAGEB10*	[4]
rs211260176	6	MARCH1	[17]
rs211260176	6	MARCH1	[17]
rs211260176	6	MARCH1	[17]
rs110128350	27	DLC1	[23]
rs109170505	22	OPN1LW	[31]
rs110876480	1	GABRR3	[20]
rs109466217	2	EF4G3	[32]
rs109416157	29	INCENP	[33]
rs43399120	4	CFTR	[25]
rs109697710	4	HIBADH	[34]
rs290117704	5	SOX5	[35]
rs42601646	5	PTPRB	[36]
rs2601646	5	PTPRR	[37]
rs11065449	8	SECISBP2	[37]
rs42749302	8	CYCL2	[36]
rs109677705	11	SPAST	[38]
rs41574912	12	KLHL1	[39]
rs41965546	21	TGFIR	[40]
rs110876480	24	GRP	[41]
rs110149073	25	PRKCB	[42]
rs42736384	26	HELLS	[42]
rs109339115	29	CATSPERR1	[24]
rs109339115	29	CAPN1	[43]

**Table 3 vetsci-09-00155-t003:** Genes associated with different semen parameters.

Associated Trait	Chromosomes	Identified Candidate Genes	Reference
semen volume per ejaculate	6	MARCH1	[17]
number of motile sperm per ejaculate	6	MARCH1	[17]
Sperm motility	6	MARCH1	[17]
Percentage of live sperm	1	SPATA16	[31]
Total number of sperms	14	RPLIOL	[31]
	1	NYD-SP5	[31]
	10	SPESP1	[31]
Semen motility	27	COX7A2L	[31]
	25	DNAH3	[44]
	5	PRP11	[45]
Ejaculate volume	10	PSMBS	[46]
	16	NR5A2	[46]
	10	PRMTS	[46]
Sperm concentration	25	FSCN1	[46]
Number of motile sperm	24	IQCJ	[46]
Number of sperms per ejaculate	3	LXH8	[46]
	24	NPC1	[46]
	8	DMRT1	[46]
EV, SPC, TSN, SM, and PTM	Y	ZNF280AY	[47]
Semen quality	Y	SOX5 (SRY-box5)	[28]

**Table 4 vetsci-09-00155-t004:** Breeds, genes, and traits of association.

Breeds	Genes	Traits	Reference
Dairy breed, chinese holstein	ETNK1, PDE3A, CSF1R, WTI, DSCAML1, SOD1, and RUNX	Semen traits	[5]
	MARCH1, PDGFRB, and PDE3A	SVPE, SCPE, NSPE, and NMSPE	[17]
	PSMS, PRMT5, ACTB, PBDE3A, FSCN1, NR5A2, IQCG, LANX8, and DMRT1	VE, SM, SC, NSP, and NMSP	[46]
	MC4R, ETNK1, LOC785875, TRIM36, ALPL, PRKCB, HIBADH, KHL1, PD2RN4, CTTR, SRD5A2, CAPN1, CATSPER1, ATP5O GABRR3 EIF4G3, SOX5 PTPRB PTPRR, SECISBP2, CYLC2, CRYZL1, LOC785875 ZBTB40, ALPL, AGBL4 ST7, PDZRN4, CNOT2, ETNK1 MRPL1, RAPGEF6, FREM1 GADD45G SPIN1 GRIN3A TRIM36 LOC511898 TGFA, FAM84B, LOC100139627, CYP2C87, SORCS1, LOC101905219, and DLC1	Poor SM	[3,32,33,34,35,36,37,38,39,40,41,42,43,44,45,46,47]
	FSHR, INHBA, INHA, and PRL	VOL, SCON, MOT, FMOT, AIR, and ASR	[29]
Brown swiss	WDR19	Semen quality	[22]
Dual purpose breeds, holstein friesian			
	DCP1, SFMBT1, GALC, PRKCD, PHF7, TLR9, SPATA7, and TMEM110	SV and TNP	[14]
	PDRK2 and GALNT13		
	MAGEB10 and KLH13	SV, MS, M, SC, and NS	[4]
Beef breeds, crossbreeds (angus, simmental, piedmontese, gelbvieh, charolais, and limousine.)	WTAP, ACAT2, TCP1, EZR, PRKN, PACRG, PLCB4, LAMP5, PAK5LMBRD2, UGT3A2, CAPSL, IL7R, SPEF2, SKP2, PRLR, and DCC	SM	[16]
American angus	HERC2, OCA2, and LOC101902976	VOL, CONC, NSP, IMot, PTMot, 3HRPTMot, %NORM, PRIM, and SEC	[18]

**Table 5 vetsci-09-00155-t005:** Genes identified in this GWAS studies and their full names.

Gene ID	Gene Full Name	Gene ID	Gene Full Name
TGFA	Transforming growth factor, Ralph [3]	PRKCB	Protein kinase C, beta [3,52]
ETNK1	Ethanolamine kinase 1 [3,5]	HIBADH	3-Hydroxyisobutyrate dehydrogenase [3,33]
LOC785875	Olfactory receptor 5K3-like [3,53]	INCENP	Inner centromere protein antigens 135/155 kDa [3,43]
ALPL	Alkaline phosphatase, liver/bone/kidney [54]	SOD1	superoxide dismutase 1 [5]
TRIM36	Tripartite motif containing 36 [3]	ACAT2	Acetyl-CoA Acetyltransferase 2 [16]
MC4R	Melanocortin 4 receptor [15]	TCP1	T-complex protein 1 subunit alpha [16]
RSPH3	Radial Spoke Head 3 [16]	GABRR3	Gamma-Aminobutyric Acid Type A Receptor Subunit Rho3 [3,20]
TAGAP	T Cell Activation RhoGTPase Activating Protein [16]	CAPN1	Calpain 1 (mu/I) large subunit [43]
FNDC1	Fibronectin Type III Domain Containing [16]	CATSPER1	Cation channel, sperm-associated 1 [3,22]
F9	Coagulation factor IX [55]	ATP5O	ATP synthase, H+ transporting, mitochondrial [3,54]
WDRD	WD Repeat Domain 19 [22]	EIF4G3	Eukaryotic translation initiation factor 4 gamma, 3 [3]
PTPRR	Protein tyrosine phosphatase, receptor type, R [3,36]	CYLC2	Cylicin, basic protein of sperm head cytoskeleton 2 [3,56]
SOX5	SRY-Box Transcription Factor 5 [3,35]	CRYZL1	Crystallin, zeta (quinone reductase)-like 1 [3]
PTPRB	Protein tyrosine phosphatase, receptor type, B [3,36]	LOC785875	olfactory receptor 5K3-like [3]
SECISBP2	SECIS binding protein 2Cylicin, basic protein of sperm head cytoskeleton 2 [37]	OCA2	oculocutaneous albinism I [57]
KLHL1	kelch-like 1 (Drosophilia) [39]	AGBL4	ATP/GTP binding protein-like 4 [3]
PDGFRB	Platelet-derived growth factor receptor beta [5,17,58]	RAPGEF6	Rap quinine nucleotide exchange factor (GEF) 6 [3]
PDE3A	phosphodiesterase 3A [5,58]	ZBTB40	Zinc finger and BTB domain containing 40 [3]
DCP1	decapping mRNA 1A [16]	ST7	Suppression of tumorigenicity7 [3]
SFMBT1	Scm-like with four mbt domains 1 [14]	CNOT2	CCR4-NOT transcription complex, subunit 2 [3]
PRKCD	protein kinase C [14]	ETNK1	Ethanolamine kinase 1 [3]
KLH13	Kelch-Like Family Member 13 [4]	MRPL1	Mitochondrial ribosomal protein L1 [3]
HERC2	HECT and RLD Domain Containing E3 Ubiquitin Protein Ligase 2 [57]	FREM1	FRAS1 related extracellular matrix1 [3]
MARCH1	membrane-associated ring finger 1 [17,59]	SPIN1	spindling 1 [3]
PHF7	PH finger protein 7 [60]	KIRREL3	Kirre-like nephrin family adhesion molecule 3 [56]
GALC	Galactosylceramidase [61]	LOC511898	Protein disulfide isomerase family A, member 6 [3]
TLR9	Toll-like receptor 9 [56]	SPATA7	spermatogenesis associated 7 [3]
TMEM110	transmembrane protein 110 [14]	NR5A2	Nuclear Receptor Subfamily 5 Group A [46]
		MAGEB10	melanoma antigen family B10 [4]

## Data Availability

There was no new data analysed for this study.

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
