# Peer review of "Candidate Genes in Bull Semen Production Traits: An Information Approach Review"

_vetsci, 2022, doi:10.3390/vetsci9040155_

Round 1

Reviewer 1 Report

Modiba et al., try to summary recent studies on candidate genes in bull semen production traits, which give some clues for the community. Anyhow, this review was not well orgnized:

(1)The author need to summary how many references in this filed in a time range, how they to analyze these reference.

(2)The manuscript should be in a more logic manner, and summarize data in different traits, and also beef and dairy.

(3) Reference format in manuscript citing is not corrent.

(4)There are some Y chromosome studies that are not in manuscript.

(5)The English need extensive revision.

Author Response

Dear Reviewer Thank you for your comment, I have learned a great deal from you and appreciate the time you took. kindly find attached my esponse to the comments. 

Thank you 

Cate Modiba 

Reviewer 2 Report

The review is carefully prepeared based on available literature on genes identified in bull semen production traits. No main concerns have been found however before publication the Authors should 

1.chcek through the manuscript the use of abreviations and/or their extenions e.g. A.I centers,PC, SV and SM, BTA, VOL, NS, and MOT, GWAS is used first as abbreviation;  SNAP is several times explained 
2.lines 34, 35 it is not clear "selecting for top"

Author Response

Dear Dr

Thank you for the comments, kindly see attached responses, they were very helpful. 

Regards 

Cate Modiba 

Reviewer 3 Report

Abstract

l 20 “The use genome wide association have shown to be successful in identify“ – Please reconstruct the sentence

l 21 “Literature have reproted“ please correct

second part of the Abstract (from line 20) needs to be rewritten

Introduction

l29  “A.I centers” define AI before use – A.I?

l34 To my knowledge, total volume and total number of spermatozoa play a crucial role in selecting bulls – bulls with low total sperm number can only be used for a low amount of semen doses and therefore they are less valuable…

l46 “Fortes et al. (2013)…“ in what species?

Association studies

l58-71 This is more an introduction to GWAS – and much too long for a review article – please shorten!

l96f “associated with five semen traits. rs41625599, rs41584616, 96 rs42012507). Thier study…” please correct the sentence

l132 “Table 2. Cndidate genes“ please correct.

l152, Table 1 (wrong number!) – first line – put the population size on the beginning and add the “breed” to the population size

l156 Although

l168 “have been identified to be associated with…”

l169/170 “showed an association”

l174 “sperm plasma integrity.“ plasma membrane integrity?

l176-183 “Hering…” please restyle this part with the two approaches

l185f “signif-icant marker can overlook another genes“ signify/cant marker can overlook other genes…

l189 “[48] and [49]“ - [48; 49]

l203 “BTA5. (Figure 4) diplict all the genes found during different studies. ndidate gene “ please correct

l222, Table 4 – please correct the style of the table!

l274 – What species does Table 5 refer to? Please mention this.

Author Response

Dear Dr. 

Thank you for comments they were very helpful in helping my manuscript 

Regards 

Cate Modiba 

Round 2

Reviewer 1 Report

I don't think the authors have addressed my concerns

Reviewer 3 Report

Much better now! Only a few suggestions:

l77 Please define “BTA” at first use

l102 “CONC” was already used as “SCONC”

There is a style change from line 150 to 152 – please correct

l153 “showed an association

l155 you used other definitions for the terms before – please unify them

l252 definition of terms again – why do you use / define these abbreviations, when you do not want to use them later? please check this.